# Design method for the relocation of plastic hinges in prefabricated steel beams with corrugated webs

**Haotian Jiang** [1]*, **Chenhua Jin**[1], **Lei Yan**[2], **QN Li**[3], **Wei Lu**[3]

**1** Jin Ling Institute of Technology, Nanjiang, China, **2** Chongqing Three Gorges University, Chongqing, China, **3** Xi'An University of Architecture and technology, Xi'an, China

* 316195906@qq.com

**Data Availability Statement:** All relevant data are within the manuscript and its Supporting Information files.

**Funding:** The authors would like to thank the Innovation Chongqing Three Gorges reservoir bank

## Abstract

The plastic hinge is a key factor in the ductile and plastic design of structures and an important basis for the seismic strengthening of structures. The formation and behavior of plastic hinges is critical for the seismic performance of an entire structure. The relocation of plastic hinges away from the beam end is an effective way of addressing brittle failure. In this paper, the cause of the shear buckling failure of prefabricated steel beams with corrugated webs and the strain variation at the flanges of steel beams are theoretically analyzed through structural tests. Based on the analysis results, a local strengthening method is proposed, and the effects of the beam sizes and the strengthening steel plate on the plastic hinges are obtained. In addition, a calculation method for determining the size of the strengthening steel plate that promotes the relocation of the plastic zone away from the beam end is given, a design method for plastic hinge relocation is proposed based on the test data, and its validity is verified.

## Introduction

Welded steel frame structures were severely damaged in the 1994 Northridge Earthquake in Los Angeles, United States and in the 1995 Kobe Earthquake in Japan, causing enormous economic losses. The core regions of beam-column joints in traditional steel frames exhibited brittle failure with extremely poor ductility during the earthquakes [1–3]. This failure mainly occurred because the joints failed to develop the ductility of the materials due to the influence of factors such as the configuration, force transfer, welds, and stress concentration in the core regions of the joints, resulting in the brittle failure of the structure prior to the formation of plastic hinges at the beam ends[4–6]. This result led to the initiation of research on the improvement of design methods for the beam-column joints of steel structures [7–9]. The relocation of the plastic hinges away from the beam end is an effective method for addressing the problem of the seismic performance of beam-column connections in steel frames while improving the constructability of the structure [10–13].

According to the requirements of the Chinese Standard "Code for Design of Steel Structures"[14], the design strength of structural members should be lower than that of the joint to

slope and engineering structure disaster prevention and control engineering Cover Letter technology research center (Grant No. SXAPGC18YB05P), Jin Ling Institute Of Technology (Grant No. jit-b-201916) and Jiangsu Province Department of Education (Grant No. 20KJD560003) support of this research work.

**Competing interests:** The authors have declared that no competing interests exist.

satisfy the design principle of "strong joint and weak member". This design strength can be achieved in two ways: by strengthening the joint or by weakening the members [14]. Both ways can relocate the critical section from the joint to the interior of the member, thereby causing the member to fail prior to the joint. As a result, the ductility of steel materials is fully used to improve the seismic performance of the joint by energy dissipation through the rotation of the plastic hinges.

## Experimental program

### The design and details of specimens

Steel beams with corrugated webs (SBCWs) were manufactured in a factory. Q235 steel was used for the flange and the corrugated webs. The flange plate has a cross section with a thickness of 10 mm and a width of 200 mm. The corrugated web has a depth of 500 mm, a single wavelength of 150 mm, an amplitude of 20 mm, and a thickness of 2.5 mm. In specimen PRCS-01, a steel plate was used to strengthen the steel beam at the joint. The strengthening steel plate uses Q235 steel and has a height of 500 mm, a length of 450 mm along the steel beam, and a thickness of 5 mm. The detailed construction is shown in Fig 1A–1C.

The steel beam in specimen PRCS-02 was manufactured by strengthening the corrugated web with diagonal bars. The diagonal bars were made by bending HRB400 rebar with a diameter of 28 mm and placing them in an X-arrangement on both sides of the corrugated web. The layout region is from the outer edge of the strengthening steel plate at the beam end to the end of the axially loaded bars of the steel beam, and the diagonal bars were welded to the flange. The construction is shown in Fig 2. The finished product is shown in Fig 3.

### Material property tests

**Loading device.** The Schematic diagram of the test loading device is shown in Fig 4.

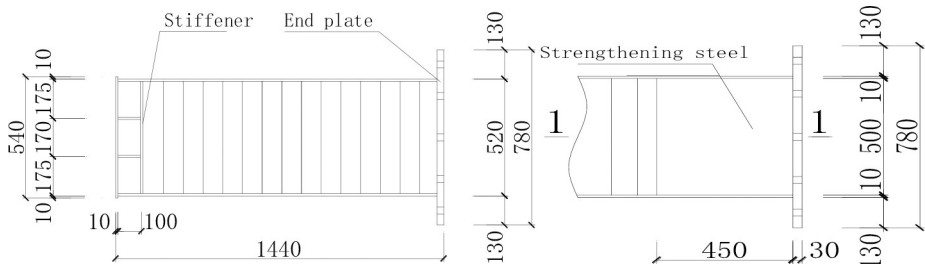

(a) PRCS-01 construction          (b) Details of local strengthening

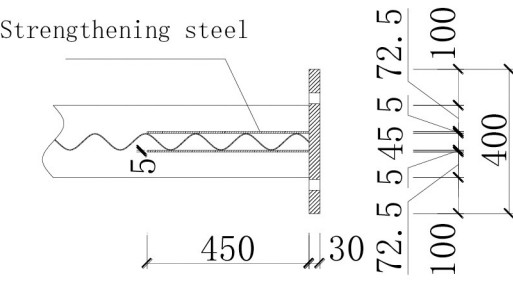

(c) 1-1 sectional view

**Fig 1. Steel frame dimensions of specimen PRCS-01.**

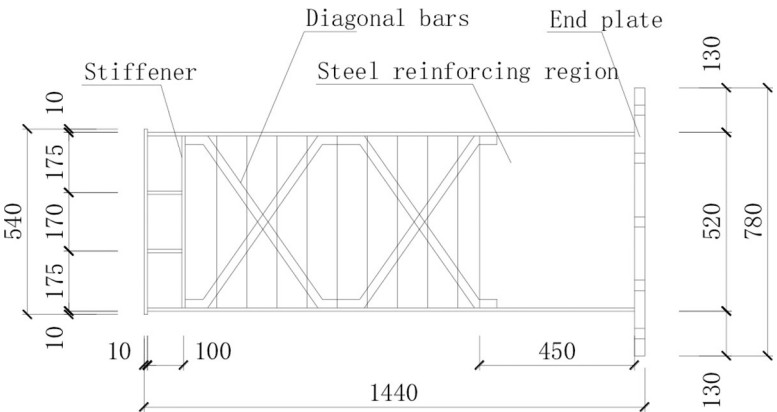

**Fig 2. Construction of the steel beam in specimen PRCS-02.**

**Loading regime.**   Prior to the test, the specimen was geometrically aligned, and a vertical load was applied to 15% of the predetermined load. While the vertical load was kept unchanged, the specimen was physically aligned, and the specimen and instruments were calibrated. After the calibration, the specimen was unloaded. Then, the specimen was loaded to the predetermined vertical load. The horizontal reciprocating load was applied in steps. The yielding of the specimen was determined by checking if the tensile strain at the top and bottom flanges of the SBCW had reached the yield strain. Before yielding, force control was adopted, and the load at each step was reciprocated once; after yielding, displacement control was adopted, the displacement at each step was equal in value to the yield displacement, and the displacement at each step was reciprocated three times. When the load dropped below 85% of the ultimate load, the specimen was considered to have failed, and the loading was stopped. During the test, the continuity and uniformity of the reciprocating loading were always ensured, and the loading or unloading rate was maintained to be consistent.

**Layout of the measuring points.**   The layout of strain gauge and displacement gauges on the steel beam is shown in Fig 5. Two strain rosettes were placed on the steel plate hoop in the joint area to the north of the column to measure the variation of the strain of the steel plate

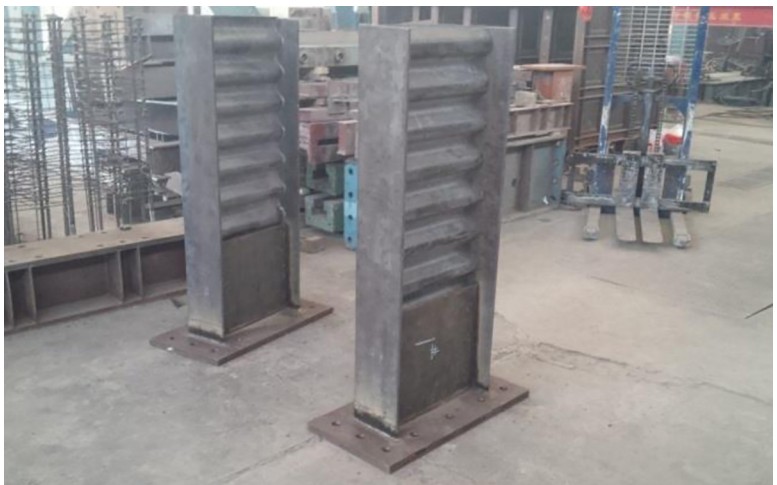

**Fig 3. The finished steel beams.**

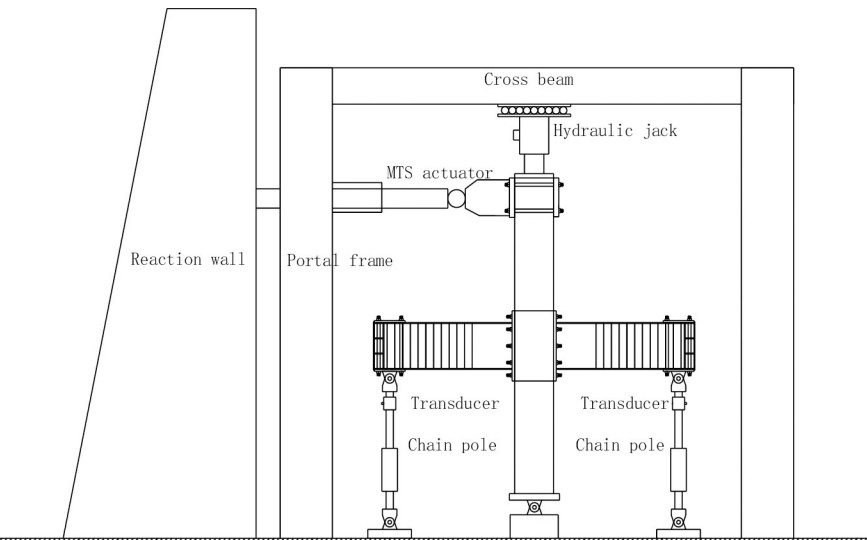

**Fig 4. Schematic diagram of the test loading device.**

hoop during the test. Strain gauges were placed on the upper and lower flanges of the east and west steel beams to measure the strain variation of the flanges due to flexure. Strain gauges and strain rosettes were placed on the strengthening steel plate at the ends of the east and west beams to measure the strain of the strengthening steel plate.

**Cracking patterns and failure modes.** During the force-controlled loading stage of specimens PRCS-01 and PRCS-02, when the specimens were subjected to reciprocating loads of

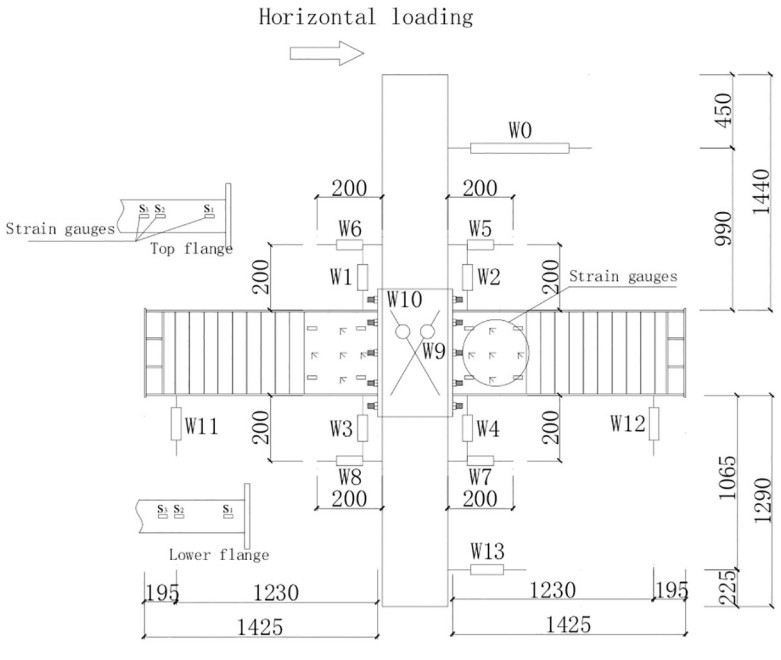

**Fig 5. Layout of the strain gauges and displacement gauges on the steel beam.**

180kN and 240kN, respectively, the top displacement of the column was 35 mm. Based on the monitoring results from the strain gauges on the specimens, the stress on the central axis of the flange away from the side of the joint at the edge of the strengthening steel plate of the SBCW reached the yield stress. Thus, the yield displacement was set to Δ = 35mm. According to the test loading regime, displacement-controlled loading was adopted in the subsequent test stage. As specimen PRCS-01 was pulled westward in the first cycle of the reciprocating displacement equal to twice the yield displacement, local buckling occurred in the middle of the corrugated web near the axially loaded bars close to the end of the east SWCB; that is, the buckling occurred 1100 mm from the central axis of the column. In comparison, as specimen PRCS-02 was subjected to the reciprocating displacement equal to the yield displacement, the existing local buckling in the corrugated web at the end of the axially loaded bars of the east and west SBCWs was aggravated and developed towards the upper and lower flanges. In particular, the wrinkles formed by the local buckling of the corrugated web of the east beam were approximately 45˚ from the flange. After the corrugated web buckled, the concentrated force at the top of the column decreased to 85% of its original value, and the test was terminated. The failure modes of specimens PRCS-01 and PRCC-02 are shown in Fig 6A and 6B.

## Failure analysis of SBCWs

To facilitate calculation and analysis, the specific values of various parameters are listed in Table 2.

According to the "Technical Specification for Steel Structures with Corrugated Webs" [15] from the China Association for Engineering Construction Standardization, the shear strength of a structural member with a corrugated web subjected to flexure in its principal plane is calculated using the following formula.

$$\tau = \frac{V}{A_{\mathrm{wn}}} \leq f_{\mathrm{v}} \tag{1}$$

Where

$V$ = the design shear acting along the web plane at the calculated cross section;

$A_{\mathrm{wn}}$ = the net cross-sectional area of the corrugated web after subtracting the openings; and

$f_{\mathrm{v}}$ = the design shear strength of the web steel.

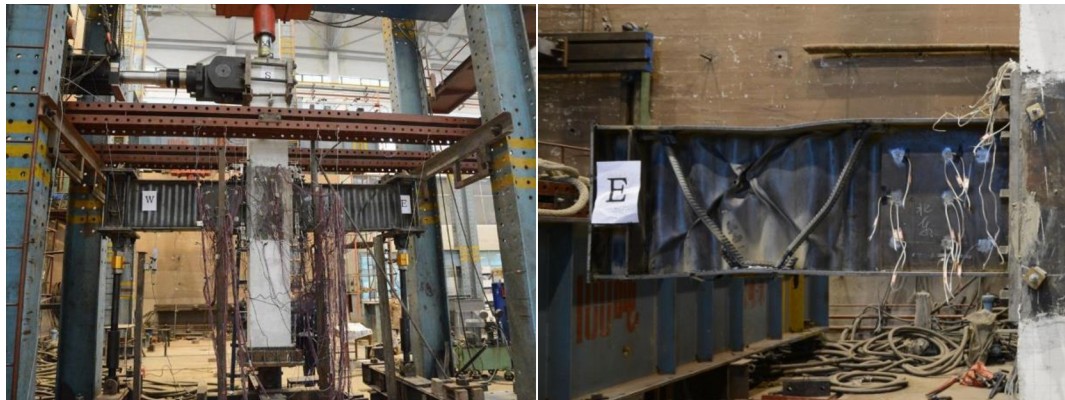

(a) Failure mode of specimen PRCS-01     (b) Local failure mode of specimen PRCS-02

**Fig 6. Local failure modes.**

**Table 1. Steel material property indicators.**

| Type | Diameter/plate thickness (mm) | Yield strength $f_y$ (N/mm$^2$) | Ultimate strength $f_u$ (N/mm$^2$) | Elastic modulus $E_c$ (N/mm$^2$) | Elongation (%) |
|---|---|---|---|---|---|
| Steel beam flange | 10 | 367.5 | 532.5 | $2.07 \times 10^5$ | 31.3 |
| Strengthening steel plate | 5 | 296.7 | 448.3 | $2.03 \times 10^5$ | 29 |

The shear capacity of the postbuckling corrugated web is calculated using the following formula.

$$\tau = \frac{V}{t_w h_w} \leq \eta \varphi_s f_v \tag{2}$$

Where

$t_w$ = the thickness of the corrugated web;

$\varphi_s$ = the stability coefficient of the shear capacity of the web, calculated as follows:

$$\varphi_s = \begin{cases} 1 - 0.35\lambda_s^2 & \lambda_s < 0.6 \\ -0.5\lambda_s^2 + 0.25\lambda_s + 0.895 & 0.6 \leq \lambda_s < 1.2 \\ 0.7/\lambda_s^2 & \lambda_s \geq 1.2 \end{cases} \tag{3}$$

$\lambda_s$ = the normalized web slenderness ratio for calculating the shear of corrugated webs, using formula (4):

$$\lambda_s = \max\left( \frac{t_w^{1/8} h_w}{173.4(q/s)^{1/8} I_{z1}^{3/8}} \sqrt{\frac{f_y}{235}}, \frac{\sqrt{40s/t_w} - 22}{85.6} \sqrt{\frac{f_y}{235}} \right) \tag{4}$$

$q$ = the wavelength of a single repeated wave of the corrugated web;

$S$ = the arc length of a single repeated wave of the corrugated web, calculated using formula (5):

$$s = q\left( 3.88\frac{a^2}{q^2} + 1.07\frac{a}{q} + 0.95 \right) \tag{5}$$

$I_{z1}$ = the out-of-plane moment of inertia per unit length of the corrugated web about the neutral axis, calculated using formula (6):

$$I_{z1} = \frac{a^2 t_w}{2}\left( 1.054 - 0.945\frac{a}{q} - 0.277\frac{a^2}{q^2} \right) \tag{6}$$

$\eta$ = the reduction factor for the shear capacity of the web with openings, with a value of 1.0 when there is no opening.

The data in Table 1 are substituted into the above formulas to obtain the parameters, as listed in Table 3.

The geometrical dimensions of the corrugated web members in Table 2 and the calculation results of the parameters in Table 3 are substituted into formula (2) to obtain the design shear

**Table 2. Parameters of SBCWs.**

| Parameters | Flange width $b_f$ (mm) | Flange thickness $t_f$ (mm) | Web depth $h_w$ (mm) | Web thickness $t_w$ (mm) | Web amplitude $a$ (mm) | Half-wavelength $\lambda$ (mm) |
|---|---|---|---|---|---|---|
| Dimensions | 200 | 10 | 500 | 2.5 | 20 | 75 |

**Table 3. Parameters for calculating the shear stability of corrugated webs.**

| Parameters | Shear stability coefficient $\varphi_s$ | Normalized web slenderness ratio $\lambda_s$ | Out-of-plane moment of inertia $I_{z1}$ (mm³) | Wave arc length $s$ (mm) |
|---|---|---|---|---|
| Calculation results | 0.955 | 0.36 | 461.54 | 174.25 |

capacity of the steel beam in the test as:

$$V = 1.49 \times 10^5 \, \text{N}$$

In view of the relation between the bending moment and the shear shown in Fig 7 and the calculation formula (7) for the plastic bending moment $M_p$ of the SBCW [13, 16, 17], the shear capacity $V$ of the corrugated web should satisfy the requirement of formula (8) in order for the plastic bending moment to be reached at the starting point of the strengthening steel plate in the SBCW.

$$M_p = b_f t_f f_y (h - t_f) \tag{7}$$

$$V > \frac{M_p}{l - l_a} \tag{8}$$

Where:
$M_p$ = the plastic bending moment of the steel beam;
$f_y$ = the yield strength of the flange material;
$h$ = the cross-section depth of the steel beam;
$l$ = the distance from the loading point to the beam end; and
$l_a$ = the length of the strengthening steel plate.
Combining formulas (7) and (8) gives the condition that the shear capacity V of the steel beam with a corrugated web should satisfy

$$V > 2.42 \times 10^5 \, \text{N}$$

The steel beam used in the test had a shear capacity of only $1.49 \times 10^5$ N, which does not satisfy the requirement of the shear capacity of SBCWs set forth in the relevant technical specification. It can be seen from formula (8) that in order for the flange to reach the plastic bending moment at the starting point of the strengthening steel plate at the beam end given the shear capacity of the SBCW, the length $l$ of the SBCW should meet the requirement given in the

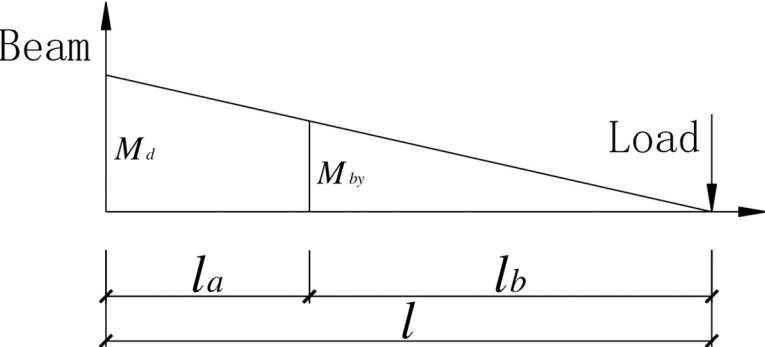

**Fig 7. The relationship between the bending moments at the starting point of strengthening at the beam end and at the beam-column joint.**

following formula.

$$l > \frac{M_{\mathrm{p}}}{V} + l_{\mathrm{a}} \qquad (9)$$

Combining formulas (7) and (9) and substituting the shear capacity $V$ of the SBCW in the test calculated from the relevant provisions of the Specification [15] into formula (9), the length $l$ of the steel beam should satisfy.

$$l > 2059\text{mm}$$

## Analysis of the relocation of the plastic zones in the SBCW

### Plastic hinge relocation

The relocation of plastic hinges can be accomplished in two ways: by joint weakening or by joint strengthening (as shown in Fig 8). Previous studies have shown that joint weakening enables the development of plasticity in beams at the cost of weakening the flange cross-section and reducing the load-carrying capacity of the beam, while joint strengthening does not reduce the load-carrying capacity of the beam and can still satisfactorily achieve the development of plasticity and good joint ductility under strong earthquakes. In this test, based on the basic principle of joint strengthening for plastic hinge relocation, the beam end was strengthened with a steel plate to achieve the relocation of the plastic zone.

### Strain analysis of the steel beam flange

The effect of a strengthening steel plate on the beam end forces was investigated to analyze the relocation of the plastic zone at the beam end in the presence of the strengthening steel plate. In the test, strain gauges were attached to the flanges of the east and west steel beams of the joint specimens to measure the variation pattern of the strain of the flanges during the entire loading and unloading processes. To facilitate a comparative analysis of the strain, the hysteresis curves of the strain with a horizontal reciprocating load were plotted. In particular, the strain variation measured by strain gauge S1 at the steel beam end and that by strain gauge S2 at the starting point of the strengthening steel plate are plotted in the same coordinate system. The strains measured at the same positions of joint specimens PRCS-01 and PRCS-02 are compared in Fig 9.

It can be seen from Fig 9 that, for the flanges of the east and west steel beam specimens, the absolute values of the strains measured by strain gage S2 are larger than those measured by the strain gage S1 during the forward and reverse loading processes. In other words, the stress of the flange at the starting point of the strengthening steel plate is higher than that of the flange at the beam end, indicating that the use of the steel plate to strengthen the beam end can relocate the plastic zone away from the beam end.

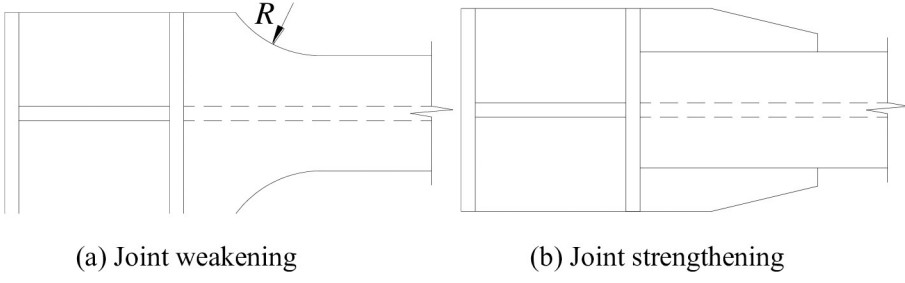

(a) Joint weakening                    (b) Joint strengthening

**Fig 8. Types of plastic hinge relocations.**

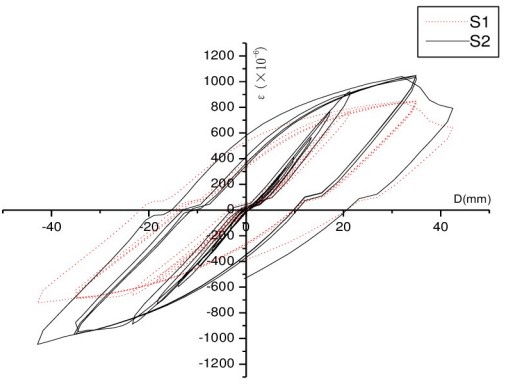
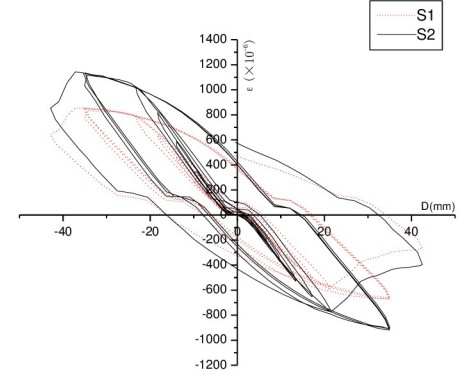

a) East beam of specimen PRCS-01 (b) West beam of specimen PRCS-01

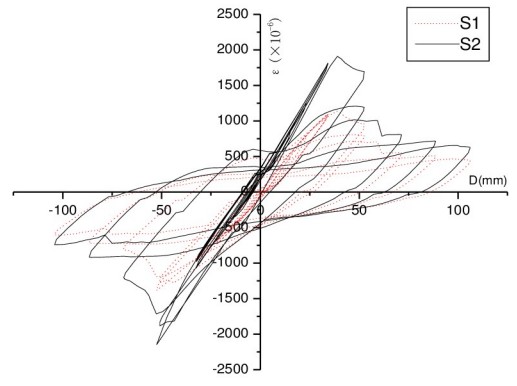
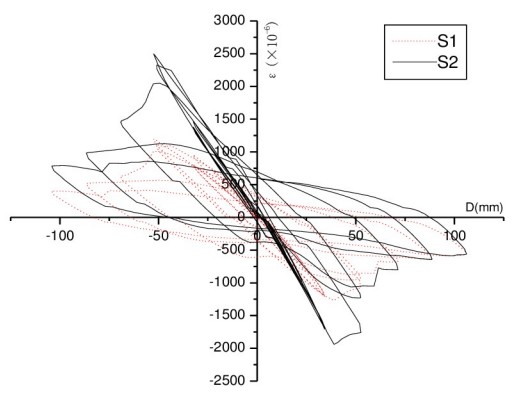

(c) East beam of specimen PRCS-02 (d) West beam of specimen PRCS-02

**Fig 9. Comparison of the strains of the flanges of the steel beam specimens.**

## Implementation of plastic zone relocation away from beam end

The beam end was strengthened using the steel plate to relocate the plastic zone away from the beam end. The strengthening steel plate was tightly and symmetrically placed against the corrugated web on its two sides and welded to the upper and lower flanges and the end plate. Fig 7 shows the relationship between the bending moment at the beam end and that on the cross section at the starting point of the strengthening region. It can be seen that in order to have the beam end and the cross section at the starting point of the strengthening region yield simultaneously, the relationship expressed in the following formula should be satisfied.

$$M_{\mathrm{dy}} = M_{\mathrm{dp}} = M_{\mathrm{p}} \cdot \frac{l}{l - l_{\mathrm{a}}} \tag{10}$$

The thickness of the strengthening steel plate $t_{\mathrm{w}}'$ can be adjusted so that the actual yield bending moment $M_{\mathrm{dy}}$ at the beam end is greater than $M_{\mathrm{dp}}$; i.e., formula (11) is satisfied, so the cross section at the starring point of the strengthening steel plate would yield earlier than that at the beam end, thereby achieving the relocation of the plastic region away from the beam

end.

$$\frac{M_{\mathrm{dy}}}{M_{\mathrm{p}}} > \frac{l}{l - l_{\mathrm{a}}} \tag{11}$$

Where

$M_{\mathrm{dy}}$ = the actual yield bending moment of the beam end;

$M_{\mathrm{dp}}$ = the actual bending moment of the beam end when the cross section at the starting point of the strengthening steel plate reaches the yield bending moment; and

$M_{\mathrm{p}}$ = the yield bending moment of the cross section at the strengthening steel plate of the steel beam.

## Design and calculation of the strengthening steel plate at the beam end

Fig 10 shows the use of the strengthening steel plate at the beam end in this test. The length $l_{\mathrm{a}}$ of the strengthening steel plate has an effect on both the relocation of the plastic zones and the ductility of the specimen. An excessively small $l_{\mathrm{a}}$ would provide insufficient space for plastic zone development and hence fail to relocate the plastic zone away from the beam end. An excessively large $l_{\mathrm{a}}$ would increase the ultimate load-carrying capacity of the specimen at the cost of decreasing its ductility. Taking into consideration the above factors, the length $l_{\mathrm{a}}$ of the strengthening steel plate is determined in reference to the range of the length of the side plate in the side-plate-strengthened joints of steel structures specified by the Architectural Institute of Japan [13].

$$l_{\mathrm{a}} = (0.5 \sim 0.75)h \tag{12}$$

The following can be obtained from formulas (7) and (1).

$$M_{\mathrm{eu}} > M_{\mathrm{dp}} = \frac{l}{l - l_{\mathrm{a}}} \cdot M_{\mathrm{p}} = \frac{l}{l - l_{\mathrm{a}}} \cdot b_{\mathrm{f}} t_{\mathrm{f}} f_{\mathrm{y}} (h - t_{\mathrm{f}}) \tag{13}$$

Where

$M_{\mathrm{eu}}$ = the ultimate flexural capacity of the steel beam end considering the postbuckling strength of the web, which can be expressed as

$$M_{\mathrm{eu}} = \gamma_{\mathrm{x}} \alpha_{\mathrm{e}} W_{\mathrm{x}} f \tag{14}$$

$$\alpha_{\mathrm{e}} = 1 - \frac{(1 - \rho) h_{\mathrm{c}}^3 t_{\mathrm{w}}}{2 I_{\mathrm{x}}} \tag{15}$$

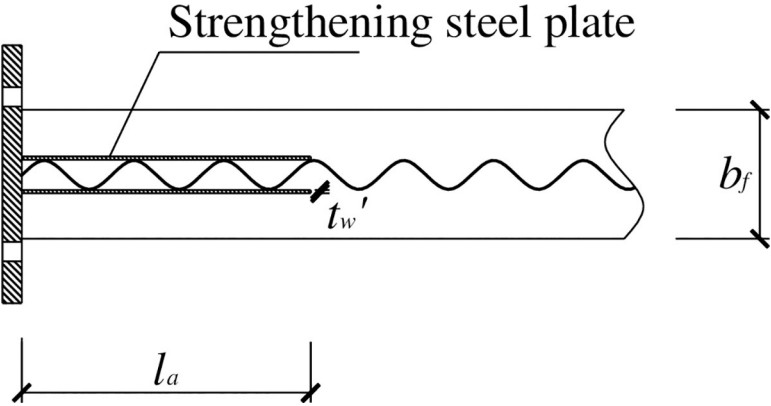

**Fig 10. Construction of the steel-plate-strengthened beam end.**

Where

$\alpha_e$ = the reduction factor for the section modulus of the beam considering the effective depth of the web;

$W_x$ = the section modulus about the $x$ axis calculated using the entire cross section of the beam;

$I_x$ = the moment of inertia about the $x$ axis calculated using the entire cross section of the beam;

$h_c$ = the depth of the compression zone of the web calculated using the entire cross section of the beam;

$\gamma_x$ = the plasticity development coefficient of the beam;

$t_w$ = the thickness of the strengthening steel plate; and

$\rho$ = the effective depth factor for the compression zone of the web.

The formula of the general slenderness ratio $\lambda_b$ for the web bending calculation is expressed as.

$$\lambda_b = \frac{h_w/t_w}{153}\sqrt{\frac{f_y}{235}} \tag{16}$$

When $\lambda_b \leq 0.85$

$$\rho = 1.0 \tag{16A}$$

When $0.85 < \lambda_b \leq 1.25$

$$\rho = 1 - 0.82(\lambda_b - 0.85) \tag{16B}$$

When $\lambda_b > 1.25$

$$\rho = \frac{1}{\lambda_b}\left(1 - \frac{0.2}{\lambda_b}\right) \tag{16C}$$

The following assumptions are made in the calculation. Before the bending moment of the flange reaches the maximum bending moment it can withstand, the strengthening steel plate at the beam end does not take any bending moment. After the bending moment of the beam end is greater than $M_p$, the strengthening steel plate resists the excess bending moment beyond the maximum bending moment that the flange can take; that is, $M_d$-$M_p$, where $M_d$ is the bending moment at the beam end. The normal bending stress of the strengthening steel plate is distributed linearly along the web. As the load increases, the normal bending stress and shear stress of the strengthening steel plate gradually increase, and the top and bottom edges of the strengthening steel plate are taken. Considering the bending stress state in which the bending stresses at the upper and lower edges of the strengthening steel plate are equal to the designed flexural strength, the bending moment that the strengthening steel plate can withstand can be calculated. By letting this bending moment equal $M_{dp}$—$M_p$, the thickness $t'$ of the strengthening steel plate can be calculated. The corresponding normal stress distribution of the strengthening steel plate [4, 18, 19] is shown in Fig 11.

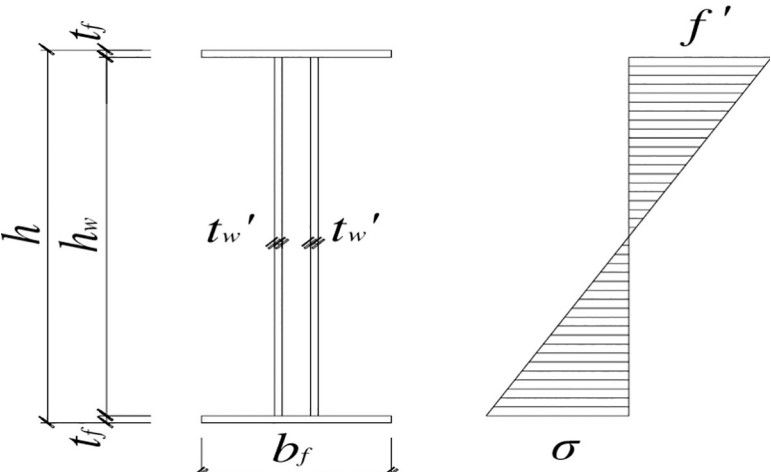

**Fig 11. Distribution of the normal stresses in the web.**

Under the stress state shown in Fig 11, the bending moment $M_t$ of the strengthening steel plate is

$$M_t = \frac{4}{9} h_w^2 t_w' f' \tag{17}$$

Letting $M_t = M_{dp} - M_p$,

$$t_w' = \frac{9 l_a}{4 h_w^2 (l - l_a) f'} \cdot M_p \tag{18}$$

By substituting formula (7) into formula (18), the thickness $t_w'$ of the strengthening steel plate can be obtained as follows.

$$t_w' = \frac{9 l_a b_f t_f (h - t_f)}{4 h_w^2 (l - l_a)} \cdot \frac{f_y}{f'} \tag{19}$$

Where
$h_w$ = the height of the strengthening steel plate;
$t_w'$ = the thickness of the strengthening steel plate;
$f'$ = the design flexural strength of the strengthening steel plate.

By substituting $t_w'$ into formula (16), the general slenderness ratio $\lambda_b$ for the flexural calculation of the strengthening steel plate is obtained. Then, the effective depth coefficient $\rho$ of the compression zone of the strengthening steel plate is calculated using the different intervals of $\lambda_b$. Finally, formula (14) is used to calculate the ultimate flexural capacity $M_{eu}$ of the steel beam end considering the postbuckling strength. The thickness $t'$ of the strengthening steel plate is adjusted to make $M_{eu}$ greater than $M_{dp}$ to achieve the relocation of the plastic zone away from the beam end. The specific values of the dimensions of the SBCW and the length of the strengthening steel plate in the test are substituted into formula (19) to calculate the thickness $t_w'$ of the strengthening steel plate to be 4.6 mm, which is needed for the relocation of the plastic zone of the SBCW in this test. Eventually, a steel plate with a thickness $t_w'$ of 5 mm was used to strengthen the beam end in the test. A comparative analysis of the strains at the beam end showed that the plastic zone was relocated away from the beam end in the test, proving

that the thickness of the strengthening steel plate was reasonably calculated using the above method.

## Conclusion

1. To avoid the brittle failure of joints in steel frames, this study investigated the relocation of the plastic hinges of steel beams and proposed two approaches: joint strengthening and joint weakening. Existing studies have shown that the joint strengthening approach can lead to plastic hinge relocation without reducing the load-carrying capacity of the steel beam to ensure good ductility of the joint. Through a comparative analysis of different types of joints with plastic hinge relocation, we proposed a new method for joint strengthening and compared the strains measured by the strain gauges on the flanges of the steel beam, proving the feasibility of this strengthening method in achieving the relocation of the plastic zone.

2. The flexural and shear capacities of the SBCW were analyzed in accordance with specifications. The SBCWs were provided by our collaborator. Due to the limitations of the test conditions, the steel beams did not meet the span requirement of "strong shear weak bending", resulting in shear failure of the SBCW during the test.

3. Because the SBCWs have better shear and buckling resistance than steel beams with flat webs, the web of the SBCW is usually designed to be high and thin. Therefore, it is recommended that the postbuckling strength of the strengthening steel plate be considered in the design calculations of the strengthening steel plate for the beam end. Because of constructional reasons, the corrugated web does not withstand any bending moment. When calculating the flexural capacity of the strengthened beam end, it is only necessary to consider the effects of the flanges and the strengthening steel plate. Based on the aforementioned analysis and in reference to current specifications and related references, the thickness of the strengthening steel plate was designed and calculated. The test results prove the validity of this calculation method, and it is expected to provide a reference for the popularization and application of this new type of joint.

## Supporting information

**S1 Data.**
(XLSX)

## Acknowledgments

Thank Professor QN Li for his guidance in writing this article. Thank Chenhua Jin, Lei Yan and Wei Lu for their contribution to this paper.

## Author Contributions

**Data curation:** Chenhua Jin.

**Formal analysis:** Haotian Jiang.

**Project administration:** Lei Yan, Wei Lu.

**Supervision:** QN Li.

**Writing – original draft:** Haotian Jiang.

**Writing – review & editing:** Haotian Jiang.

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
