## [Decision Letter · Decision Letter 0]

31 Dec 2020

PONE-D-20-32421

Design method for the relocation of plastic hinges in prefabricated steel beams with corrugated webs

PLOS ONE

Dear Dr. JIANG,

Thank you for submitting your manuscript to PLOS ONE. After careful consideration, we feel that it has merit but does not fully meet PLOS ONE’s publication criteria as it currently stands. Therefore, we invite you to submit a revised version of the manuscript that addresses the points raised during the review process.

Comments from two reviewers are attached. Some improvements are still required, particularly the latest literature within 3-5 years should be reviewed and commented.The novelty should be clearly highlighted through literature review.

We look forward to receiving your revised manuscript.

Kind regards,

Jianguo Wang, PhD

Academic Editor

PLOS ONE

Journal Requirements:

"The authors would like to thank the Innovation Chongqing Three Gorges reservoir

bank slope and engineering structure disaster prevention and control engineering

technology research center (Grant No. SXAPGC18YB05P) and Jin Ling Institue Of

Technology (No.jit-b-201916)support of this research work."

"Haotian Jiang conceived the study and wrote the paper. Jin Chenhua carried out

the experiments and data analysis. Lei Yan and Lu Wei led the projects and reviewed the manuscript."

5. Please amend the manuscript submission data (via Edit Submission) to include authors Jin Chenhua,Yan Lei.

6. We note you have included a table to which you do not refer in the text of your manuscript. Please ensure that you refer to Tables 3, 4, 5, 8 in your text; if accepted, production will need this reference to link the reader to the Table.

Reviewers' comments:

Reviewer's Responses to Questions

**Comments to the Author**

1. Is the manuscript technically sound, and do the data support the conclusions?

Reviewer #1: Yes

Reviewer #2: Yes

2. Has the statistical analysis been performed appropriately and rigorously? 

Reviewer #1: Yes

Reviewer #2: Yes

3. Have the authors made all data underlying the findings in their manuscript fully available?

Reviewer #1: Yes

Reviewer #2: Yes

4. Is the manuscript presented in an intelligible fashion and written in standard English?

Reviewer #1: Yes

Reviewer #2: Yes

5. Review Comments to the Author

Reviewer #1: 1. How to ensure that the component is destroyed before the node?

2. Q235 steel, why not choose high strength steel?

3. The length of steel beam calculated in this paper should be greater than 2059 mm, but the actual length is 1440 mm. How to meet the shear requirements?

4. If the reinforced steel plate is welded on the flange and end plate, will the weld formed by welding cause stress concentration phenomenon on the component itself,

5. If this phenomenon occurs, should the bearing capacity and ductility of members be considered?

Reviewer #2: 1. The Literature Review is not adequate. Please refer the following articles:

[1] Wei Li, Linzhu Sun*, Junliang Zhao, Pengfei Lu, Fang Yang. Seismic Performance of Reinforced Concrete Columns Confined with Two Layers of Stirrups. The Structural Design of Tall and Special Buildings, 2018, 27(12):1279-1291, https://doi.org/10.1002/tal.1484

[2] Sun LZ, Li W*. Cyclic behavior of reinforced concrete columns confined with two layers of stirrups. Structural Concrete, 20(4): 1279-1291, 2019, https://doi.org/10.1002/suco.201800229

[3] Sun LZ, Wei ZZ, Li W*, Xiang JH. Seismic behaviour of cross-shaped steel fibre reinforced concrete columns. Archives of Civil and Mechanical Engineering, 20, 126 (2020). https://doi.org/10.1007/s43452-020-00118-x

[4] Wei Li, Jingang Xiong, Linjie Wu, Kejia Yang. Experimental study and numerical analysis on seismic behavior of composite RCS frames. Structural Concrete, https://doi: 10.1002/suco.201900068, 2020.

[5] Yang KJ, Dai T, Li W*, Hung B. Experimental investigation of seismic behavior of confined cavity walls retrofitted with reactive powder concrete overlays. Structural Concrete, 21(1): 184-198, 2020，https://doi.org/10.1002/suco.201800243

2. 0 overview is suggested revised as”0 introduction”.

3. 1 Experimental scheme should be revised as“ Experimental program“.

4. 1.1 Component design and dimensions should be revised as“The design and details of specimens “.

5. In table 1, the stress-strain curves of steel should be given.

6. 1.5 Test phenomenon should be revised as“ Cracking patterns and failure modes “.

7. The English level should be improved by language editor.

6. PLOS authors have the option to publish the peer review history of their article (what does this mean?). If published, this will include your full peer review and any attached files.

Reviewer #1: No

Reviewer #2: No

---

## [Author Response · Author response to Decision Letter 0]

13 Jan 2021

Chen Maili ，e-mail：cml3635@163.com .Professor. Yan ’An University ；

Li Wei , e-mail：liweiwoaini521@hotmail.com.Professor. Wen Zhou University；

---

## [Decision Letter · Decision Letter 1]

20 Jan 2021

Design method for the relocation of plastic hinges in prefabricated steel beams with corrugated webs

PONE-D-20-32421R1

Dear Dr. JIANG,

We’re pleased to inform you that your manuscript has been judged scientifically suitable for publication and will be formally accepted for publication once it meets all outstanding technical requirements.

Kind regards,

Jianguo Wang, PhD

Academic Editor

PLOS ONE

Additional Editor Comments (optional):

Reviewers' comments:

Reviewer's Responses to Questions

**Comments to the Author**

1. If the authors have adequately addressed your comments raised in a previous round of review and you feel that this manuscript is now acceptable for publication, you may indicate that here to bypass the “Comments to the Author” section, enter your conflict of interest statement in the “Confidential to Editor” section, and submit your "Accept" recommendation.

Reviewer #2: All comments have been addressed

2. Is the manuscript technically sound, and do the data support the conclusions?

Reviewer #2: Yes

3. Has the statistical analysis been performed appropriately and rigorously? 

Reviewer #2: Yes

4. Have the authors made all data underlying the findings in their manuscript fully available?

Reviewer #2: Yes

5. Is the manuscript presented in an intelligible fashion and written in standard English?

Reviewer #2: Yes

6. Review Comments to the Author

Reviewer #2: The authors have been addressed all comments. Therefore, I suggest this paper should be accepted in this form.

7. PLOS authors have the option to publish the peer review history of their article (what does this mean?). If published, this will include your full peer review and any attached files.

Reviewer #2: No

---

## [Editor Report · Acceptance letter]

9 Feb 2021

PONE-D-20-32421R1 

Design method for the relocation of plastic hinges in prefabricated steel beams with corrugated webs 

Dear Dr. Jiang:

I'm pleased to inform you that your manuscript has been deemed suitable for publication in PLOS ONE. Congratulations! Your manuscript is now with our production department. 

Kind regards, 

on behalf of

Dr. Jianguo Wang 

Academic Editor

PLOS ONE